# Decomposing Mutual Information for Representation Learning

## Abstract

Many self-supervised representation learning methods maximize mutual information (MI) across views. In this paper, we transform each view into a set of subviews and then decompose the original MI bound into a sum of bounds involving conditional MI between the subviews. E.g., given two views $x$ and $y$ of the same input example, we can split $x$ into two subviews, $x'$ and $x''$, which depend only on $x$ but are otherwise unconstrained. The following holds: $I(x; y) \geq I(x''; y) + I(x'; y|x'')$, due to the chain rule and information processing inequality. By maximizing both terms in the decomposition, our approach explicitly rewards the encoder for any information about $y$ which it extracts from $x''$, and for information about $y$ extracted from $x'$ in excess of the information from $x''$. We provide a novel contrastive lower bound on conditional MI, that relies on sampling contrast sets from $p(y|x'')$. By decomposing the original MI into a sum of increasingly challenging MI bounds between sets of increasingly informed views, our representations can capture more of the total information shared between the original views. We empirically test the method in a vision domain and for dialogue generation.

## 1 Introduction

The ability to extract actionable information from data in the absence of explicit supervision seems to be a core prerequisite for building systems that can, for instance, learn from few data points or quickly make analogies and transfer to other tasks. Approaches to this problem include generative models (Hinton, 2012; Kingma & Welling, 2014) and self-supervised representation learning approaches, in which the objective is not to maximize likelihood, but to formulate a series of (label-agnostic) tasks that the model needs to solve through its representations (Noroozi & Favaro, 2016; Devlin et al., 2019; Gidaris et al., 2018; Hjelm et al., 2019). Self-supervised learning includes successful models leveraging contrastive learning, which have recently attained comparable performance to their fully-supervised counterparts (Bachman et al., 2019; Chen et al., 2020a).

Many self-supervised learning methods train an encoder such that the representations of a pair of views $x$ and $y$ derived from the same input example are more similar to each other than to representations of views sampled from a contrastive negative sample distribution, which is usually the marginal distribution of the data. For images, different views can be built using random flipping, color jittering and cropping (Bachman et al., 2019; Chen et al., 2020a). For sequential data such as conversational text, the views can be past and future utterances in a given dialogue. It can be shown that these methods maximize a lower bound on mutual information (MI) between the views, $I(x; y)$, w.r.t. the encoder, i.e. the InfoNCE bound (Oord et al., 2018). One significant shortcoming of this approach is the large number of contrastive samples required, which directly impacts the total amount of information which the bound can measure (McAllester & Stratos, 2018; Poole et al., 2019).

In this paper, we consider creating subviews of $x$ by removing information from it in various ways, e.g. by masking some pixels. Then, we use representations from less informed subviews as a source of hard contrastive samples for representations from more informed subviews. For example, in Fig. 1, one can mask a pixel region in $x'$ to obtain $x''$ and ask (the representation of) $x''$ to be closer to $y$ than to random images of the corpus, and for $x'$ to be closer to $y$ than to samples from $p(y|x'')$. This corresponds to *decomposing* the MI between $x$ and $y$ into $I(x; y) \geq I(x''; y) + I(x'; y|x'')$. The conditional MI measures the information about $y$ that the model has gained by looking at $x'$ beyond the information already contained in $x''$. In Fig. 1 (*left*), standard contrastive approaches

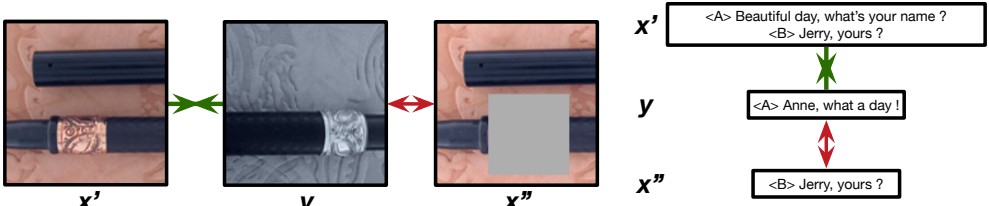

Figure 1: A demonstration of our approach in vision (*left*) and dialogue (*right*). (*left*) Given two augmentations $x$ and $y$, we fork $x$ into two subviews, $x'$ which is an exact copy of $x$ and $x''$, an information-restricted view obtained by occluding some of the pixels in $x'$. We can maximize $I(x; y) \geq I(x''; y) + I(x'; y|x'')$ using a contrastive bound by training $x''$ to be closer to $y$ than to other images from the corpus, *and* by training $x'$ to be closer to $y$ than to samples from $p(y|x'')$, i.e. we can use $x''$ to generate *hard negative* samples for $x'$. The conditional MI term encourages the encoder to imbue the representation of $x'$ with information it shares with $y$ beyond the information already in $x''$. (*right*) $x'$ and $y$ represent past and future in a dialogue respectively and $x''$ is the "recent past". In this context, the encoder is encouraged to capture *long-term dependencies* that cannot be explained by the most recent utterances.

could focus on the overall "shape" of the object and would need many negative samples to capture other discriminative features. In our approach, the model is more directly encouraged to capture these additional features, e.g. the embossed detailing. In the context of predictive coding on sequential data such as dialogue, by setting $x''$ to be the most recent utterance (Fig. 1, *right*), the encoder is directly encouraged to capture *long-term dependencies* that cannot be explained by $x''$. We formally show that, by such decomposition, our representations can potentially capture more of the total information shared between the original views $x$ and $y$.

Maximizing MI between multiple views can be related to recent efforts in representation learning, amongst them AMDIM (Bachman et al., 2019), CMC (Tian et al., 2019) and SwAV (Caron et al., 2020). However, these models maximize the sum of MIs between views $I(\{x', x''\}; y) = I(x''; y) + I(x'; y)$. E.g., in Bachman et al. (2019), $x'$ and $x''$ could be global and local representations of an image, and in Caron et al. (2020), $x'$ and $x''$ could be the views resulting from standard cropping and the aggressive multi-crop strategy. This equality is only valid when the views $x'$ and $x''$ are statistically independent, which usually does not hold. Instead, we argue that a better decomposition is $I(\{x', x''\}; y) = I(x''; y) + I(x'; y|x'')$, which always holds. Most importantly, the conditional MI term encourages the encoder to capture more non-redundant information across views.

To maximize our proposed decomposition, we present a novel lower bound on conditional MI in Section 3. For the conditional MI maximization, we give a computationally tractable approximation that adds minimal overhead. In Section 4, we first show in a synthetic setting that decomposing MI and using the proposed conditional MI bound leads to capturing more of the ground-truth MI. Finally, we present evidence of the effectiveness of the method in vision and in dialogue generation.

## 2 PROBLEM SETTING

The maximum MI predictive coding framework (McAllester, 2018; Oord et al., 2018; Hjelm et al., 2019) prescribes learning representations of input data such that they maximize MI. Estimating MI is generally a hard problem that has received a lot of attention in the community (Kraskov et al., 2004; Barber & Agakov, 2003). Let $x$ and $y$ be two random variables which can generally describe input data from various domains, e.g. text, images or sound. We can learn representations of $x$ and $y$ by maximizing the MI of the respective features produced by encoders $f, g : \mathcal{X} \to \mathbb{R}^d$, which by the data processing inequality, is bounded by $I(x; y)$:

$$\arg\max_{f,g} I(f(x); g(y)) \leq I(x; y). \tag{1}$$

We assume that the encoders can be shared, i.e. $f = g$. The optimization in Eq. 1 is challenging but can be lower-bounded. Our starting point is the recently proposed InfoNCE lower bound on MI (Oord et al., 2018) and its application to self-supervised learning for visual representations (Bachman

et al., 2019; Chen et al., 2020a). In this setting, $x$ and $y$ are paired input images, or independently-augmented copies of the same image. These are encoded using a neural network encoder which is trained such that the representations of the two image copies are closer to each other in the embedding space than to other images drawn from the marginal distribution of the corpus. This can be viewed as a *contrastive* estimation of the MI (Oord et al., 2018). We present the InfoNCE bound next.

## 2.1 INFONCE BOUND

InfoNCE (Oord et al., 2018) is a lower-bound on $I(x; y)$ obtained by comparing pairs sampled from the joint distribution $x, y_1 \sim p(x, y)$ to a set of negative samples, $y_{2:K} \sim p(y_{2:K}) = \prod_{k=2}^{K} p(y_k)$, also called *contrastive*, independently sampled from the marginal:

$$I_{NCE}(x; y|E, K) = \mathbb{E}_{p(x,y_1)p(y_{2:K})} \left[ \log \frac{e^{E(x,y_1)}}{\frac{1}{K} \sum_{k=1}^{K} e^{E(x,y_k)}} \right] \leq I(x, y), \tag{2}$$

where $E$ is a critic assigning a real valued score to $x, y$ pairs. We provide an exact derivation for this bound in the Appendix[1]. For this bound, the optimal critic is the log-odds between the conditional distribution $p(y|x)$ and the marginal distribution of $y$, $E^*(x, y) = \log \frac{p(y|x)}{p(y)} + c(x)$ (Oord et al., 2018; Poole et al., 2019). The InfoNCE bound is loose if the true mutual information $I(x; y)$ is larger than $\log K$. In order to overcome this difficulty, recent methods either train with large batch sizes (Chen et al., 2020a) or exploit an external memory of negative samples in order to reduce memory requirements (Chen et al., 2020b; Tian et al., 2020). These methods rely on uniform sampling from the training set in order to form the contrastive sets. For further discussion of the limits of variational bounds of MI, see McAllester & Stratos (2018).

## 3 DECOMPOSING MUTUAL INFORMATION

By the data processing inequality: $I(x; y) \geq I(\{x_1, \ldots, x_N\}; y)$, where $\{x_1, \ldots, x_N\}$ are different subviews of $x$ – i.e., views derived from $x$ without adding *any* exogenous information. For example, $\{x_1, \ldots, x_N\}$ can represent exchanges in a longer dialog $x$, sentences in a document $x$, or different augmentations of the same image $x$. Equality is obtained when the set of subviews retains all information about $x$, e.g. if $x$ is in the set.

Without loss of generality, we consider the case $N = 2$, $I(x; y) \geq I(\{x', x''\}; y)$, where $\{x', x''\}$ indicates two subviews derived from the original $x$. We can apply the chain rule for MI:

$$I(x; y) \geq I(\{x', x''\}; y) = I(x''; y) + I(x'; y|x''), \tag{3}$$

where the equality is obtained if and only if $I(x; y|\{x', x''\}) = 0$, i.e. $x$ doesn't give any information about $y$ in excess to $\{x', x''\}$[2]. This suggests that we can maximize $I(x; y)$ by maximizing each of the MI terms in the sum. The conditional MI term can be written as:

$$I(x'; y|x'') = \mathbb{E}_{p(x',x'',y)} \left[ \log \frac{p(y|x', x'')}{p(y|x'')} \right]. \tag{4}$$

This conditional MI is different from the unconditional MI, $I(x'; y)$, insofar it measures the amount of information shared between $x'$ and $y$ which cannot be explained by $x''$. Note that the decomposition holds for arbitrary partitions of $x', x''$, e.g. $I(\{x', x''\}; y) = I(x'; y) + I(x''; y|x')$.

When $\mathcal{X}$ is high-dimensional, the amount of mutual information between $x$ and $y$ will potentially be larger than the amount of MI that $I_NCE$ can measure given computational constraints associated with large $K$ and the poor log scaling properties of the bound. The idea that we put forward is to split the total MI into a sum of MI terms of smaller magnitude, thus for which $I_{NCE}$ would have less bias for any given $K$, and estimate each of those terms in turn. The resulting decomposed bound can be written into a sum of unconditional and conditional MI terms:

$$I_{NCES}(x; y) = I_{NCE}(x''; y) + I_{CNCE}(x'; y|x'') \leq I(x; y), \tag{5}$$

---

[1]The derivation in Oord et al. (2018) presented an approximation and therefore was not properly a bound. An alternative, exact derivation of the bound can be found in Poole et al. (2019).

[2]For a proof of this fact, it suffices to consider $I(\{x, x', x''\}; y) = I(x; y|\{x', x''\}) + I(\{x', x''\}; y)$, given that $I(\{x, x', x''\}; y) = I(x; y)$, equality is obtained iff $I(x; y|\{x', x''\}) = 0$.

where $I_{CNCE}$ is a lower-bound on conditional MI and will be presented in the next section. Both conditional (Eq. 6) and unconditional bounds on the MI (Eq. 14) can capture at most $\log K$ nats of MI. Therefore, the bound that arises from the decomposition of the MI in Eq. 5 potentially allows to capture up to $N \log K$ nats of MI in total, where $N$ is the number of subviews used to describe $x$. This shows that measuring mutual information by decomposing it in a sequence of estimation problems potentially allows to capture more nats of MI than with the standard $I_{NCE}$, which is bounded by $\log K$.

## 4 CONTRASTIVE BOUNDS ON CONDITIONAL MUTUAL INFORMATION

One of the difficulties in computing the decomposed bound is measuring the conditional mutual information. In this section, we provide bounds and approximations of this quantity. First, we show that we can readily extend InfoNCE.

**Proposition 1** (**Conditional InfoNCE**). *The following is a lower-bound on the conditional mutual information $I(x'; y|x'')$ and verifies the properties below:*

$$I_{CNCE}(x'; y|x'', E, K) = \mathbb{E}_{p(x', x'', y_1) p(y_{2:K}|x'')} \left[ \log \frac{e^{E(x'', x', y_1)}}{\frac{1}{K} \sum_{k=1}^{K} e^{E(x'', x', y_k)}} \right] \tag{6}$$

1. $I_{CNCE} \leq I(x'; y|x'')$.

2. $E^* = \arg\sup_E I_{CNCE} = \log \frac{p(y|x'', x')}{p(y|x'')} + c(x', x'')$.

3. *When $K \to \infty$ and $E = E^*$, we recover the true conditional MI:*
   $\lim_{K \to \infty} I_{CNCE}(x'; y|x'', E^*, K) = I(x'; y|x'')$.

The proof can be found in Sec. A.2 and follows closely the derivation of the InfoNCE bound by applying a result from Barber & Agakov (2003) and setting the proposal distribution of the variational approximation to $p(y|x'')$. An alternative derivation of this bound was also presented in parallel in Foster et al. (2020) for optimal experiment design. Eq. 6 shows that a lower bound on the conditional MI can be obtained by sampling contrastive sets from the proposal distribution $p(y|x'')$. Indeed, since we want to estimate the MI conditioned on $x''$, we should allow our contrastive distribution to condition on $x''$. Note that $E$ is now a function of three variables.

Computing Eq. 6 requires access to a large number of samples from $p(y|x'')$, which is unknown and usually challenging to obtain. In order to overcome this, we propose two solutions.

### 4.1 VARIATIONAL APPROXIMATION

The next proposition shows that it is possible to obtain a bound on the conditional MI by approximating the unknown conditional distribution $p(y|x'')$ with a variational distribution $\tau(y|x'')$.

**Proposition 2** (**Variational $I_{CNCE}$**). *For any variational approximation $\tau(y|x'')$ in lieu of $p(y|x'')$,*

$$I_{VAR}(x', y|x'', E, \tau, K) = \mathbb{E}_{p(x', x'', y_1) \tau(y_{2:K}|x'')} \left[ \log \frac{e^{E(x'', x', y_1)}}{\frac{1}{K} \sum_{k=1}^{K} e^{E(x'', x', y_k)}} \right] \tag{7}$$

$$- \mathbb{E}_{p(x'')} \left[ KL\left(p(y|x'') \| \tau(y|x'')\right) \right],$$

*with $p(\cdot|x'') << \tau(\cdot|x'')$ for any $x''$, we have the following properties:*

1. $I_{VAR} \leq I(x'; y|x'')$.

2. *If $\tau(y|x'') = p(y|x'')$, $I_{VAR} = I_{CNCE}$.*

3. $\lim_{K \to \infty} \sup_E I_{VAR}(x'; y|x'', E, \tau, K) = I(x'; y|x'')$.

See Sec. A.3 for the proof. This bound side-steps the problem of requiring access to an arbitrary number of contrastive samples from the unknown $p(y|x'')$ by i.i.d. sampling from the known and

tractable $\tau(y|x'')$. We prove that as the number of examples goes to $\infty$, optimizing the bound w.r.t. $E$ converges to the true conditional MI. Interestingly, this holds true for any value of $\tau$, though the choice of $\tau$ will most likely impact the convergence rate of the estimator.

Eq. 3 is superficially similar to the ELBO (evidence lower bound) objective used to train VAEs (Kingma & Welling, 2014), where $\tau$ plays the role of the approximate posterior (although the KL direction in the ELBO is inverted). This parallel suggests that $\tau^*(y|x'') = p(y|x'')$ may not be the optimal solution for some values of $K$ and $E$. However, we see trivially that if we ignore the dependency of the first expectation term on $\tau$ and only optimize $\tau$ to minimize the KL term, then it is guaranteed that $\tau^*(y|x) = p(y|x'')$, for any $K$ and $E$. Thus, by the second property in Proposition 2, optimizing $I_{VAR}(E, \tau^*, K)$ w.r.t $E$ will correspond to optimizing $I_{CNCE}$.

In practice, the latter observation significantly simplifies the estimation problem as one can minimize a Monte-Carlo approximation of the KL divergence w.r.t $\tau$ by standard supervised learning: we can efficiently approximate the KL by taking samples from $p(y|x'')$. Those can be directly obtained by using the joint samples from $p(x, y)$ included in the training set and computing $x''$ from $x$.[3]

## 4.2 Importance Sampling Approximation

Maximizing $I_{VAR}$ can still be challenging as it requires estimating a distribution over potentially high-dimensional inputs. In this section, we provide an importance sampling approximation of $I_{CNCE}$ that bypasses this issue.

We start by observing that the optimal critic for $I_{NCE}(x''; y|E, K)$ is $\bar{E}(x'', y) = \log \frac{p(y|x'')}{p(y)} + c(x'')$, for any $c$. Assuming we have appropriately estimated $\bar{E}(x'', y)$, it is possible to use importance sampling to produce approximate samples from $p(y|x'')$. This is achieved by first sampling $y'_{1:M} \sim p(y)$ and resampling $K \le M$ ($K > 0$) examples i.i.d. from the normalized importance distribution $q_{SIR}(y_k) = w_k \delta(y_k \in y'_{1:M})$, where $w_k = \frac{\exp \bar{E}(x'', y_k)}{\sum_{m=1}^{M} \exp \bar{E}(x'', y_m)}$. This process is also called "sampling importance resampling" (SIR). As $M/K \to \infty$, it is guaranteed to produce samples from $p(y|x'')$ (Rubin, 1987). The SIR estimator is written as:

$$I_{SIR}(x', y|x'', E, K) = \mathbb{E}_{p(x'', x', y_1)p(y'_{1:M})q_{SIR}(y_{2:K})} \left[ \frac{1}{K} \log \frac{e^{E(x'', x', y_1)}}{\sum_{k=1}^{K} e^{E(x'', x', y_k)}} \right], \quad (8)$$

where we note the dependence of $q_{SIR}$ on $w_k$ and hence $\bar{E}$. SIR is known to increase the variance of the estimator (Skare et al., 2003) and is wasteful given that only a smaller set of $K$ examples are actually used for MI estimation. Hereafter, we provide a cheap approximation of the SIR estimator.

The key idea is to rewrite the contribution of the negative samples in the denominator of Eq. 8 as an average $(K-1) \sum_{k=2}^{K} \frac{1}{K-1} e^{E(x'', x', y_k)}$ and use the normalized importance weights $w_k$ to estimate that term under the resampling distribution. We hypothesize that this variant has less variance as it does not require the additional resampling step. The following proposition shows that as the number of negative examples goes to infinity, the proposed approximation converges to the true value of the conditional MI.

**Proposition 3 (Importance Sampling $I_{CNCE}$).** *The following approximation of $I_{SIR}$:*

$$I_{IS}(x', y|x'', E, K) = \mathbb{E}_{p(x'', x', y_1)p(y_{2:K})} \log \frac{e^{E(x'', x', y_1)}}{\frac{1}{K} \left( e^{E(x'', x', y_1)} + (K-1) \sum_{k=2}^{K} w_k e^{E(x'', x', y_k)} \right)}, \quad (9)$$

*where $w_k = \frac{\exp \bar{E}(x'', y_k)}{\sum_{k=2}^{K} \exp \bar{E}(x'', y_k)}$ and $\bar{E} = \arg\sup_E I_{NCE}(x'', y|E, K)$, verifies:*

1. $\lim_{K \to \infty} \sup_E I_{IS}(x'; y|x'', E, K) = I(x'; y|x'')$,

2. $\lim_{K \to \infty} \arg\sup_E I_{IS} = \log \frac{p(y|x'', x')}{p(y|x'')} + c(x', x'')$.

---

[3]The ability to perform that computation is usually a key assumption in self-supervised learning approaches.

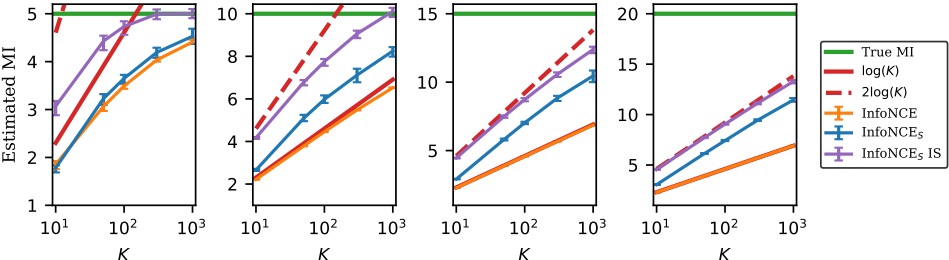

Figure 2: We plot the value of the MI estimated by $I_{NCE}$ and $I_{NCES}$ bounds for three Gaussian covariates $x', x'', y$ as function of the number of negative samples $K$. We sample different covariances for a fixed true MI (green horizontal line) and report error bars. "InfoNCE" computes $I_{NCE}(x', x''; y)$; "InfoNCE$_S$" computes $I_{NCE}(x''; y) + I_{CNCE}(x'; y|x'')$; "InfoNCE$_S$ IS" computes $I_{NCE}(x''; y) + I_{IS}(x'; y|x'')$.

The proof can be found in Sec. A.4. This objective up-weights the negative contribution to the normalization term of examples that have high probability under the resampling distribution. This approximation is cheap to compute given that the negative samples still initially come from the marginal distribution $p(y)$ and avoids the need for resampling. The proposition shows that in the limit of $K \to \infty$, optimizing $I_{IS}$ w.r.t. $E$ converges to the conditional MI and the optimal $E$ converges to the optimal $I_{CNCE}$ solution. We also note that we suppose $E$ The $I_{IS}$ approximation provides a general, grounded way of sampling "harder" negatives by filtering samples from the easily-sampled marginal $p(y)$.

## 5 EXPERIMENTS

We start by investigating whether maximizing the decomposed MI using our conditional MI bound leads to a better estimate of the ground-truth MI in a synthetic experiment. Then, we experiment on a self-supervised image representation learning domain. Finally, we explore an application to natural language generation in a sequential setting, such as conversational dialogue.

### 5.1 SYNTHETIC DATA

We extend Poole et al. (2019)'s two variable setup to three variables. We posit that $\{x', x'', y\}$ are three Gaussian co-variates, $x', x'', y \sim \mathcal{N}(0, \Sigma)$ and we choose $\Sigma$ such that we can control the total mutual information $I(\{x', x''\}; y)$ such that $I = \{5, 10, 15, 20\}$ (see Appendix for pseudo-code and details of the setup). We aim to estimate the total MI $I(\{x', x''\}; y)$ and compare the performance of our approximators in doing so. For more details of this particular experimental setting, see App. B.

In Figure 2, we compare the estimate of the MI obtained by:

1. *InfoNCE*, which computes $I_{NCE}(\{x', x''\}, y|E, K)$ and will serve as our baseline;

2. *InfoNCEs*, which probes the effectiveness of decomposing the total MI into a sum of smaller terms and computes $I_{NCE}(x'', y|E, K/2) + I_{CNCE}(x', y|x'', E, K/2)$, where $K/2$ samples are obtained from $p(y)$ and $K/2$ are sampled from $p(y|x'')$;

3. *InfoNCEs IS*, the decomposed bound using our importance sampling approximation to the conditional MI $I_{IS}$, i.e. $I_{NCE}(x'', y|E, K) + I_{IS}(x', y|x'', E, K)$. This does not require access to samples from $p(y|x'')$ and aims to test the validity of our approximation in an empirical setting. Both terms reuse the same number of samples $K$.

For 2., we use only half as many samples as InfoNCE to estimate each term in the MI decomposition ($K/2$), so that the total number of negative samples is comparable to InfoNCE. Note that we use $K$ samples in "InfoNCE IS", because those are reused for the conditional MI computation. All critics $E$ are parametrized by MLPs as explained in Sec. B. Our results in Figure 2 show that, for larger amounts of true MI, decomposing MI as we proposed can capture more nats than InfoNCE with an order magnitude less examples. We also note that the importance sampling estimator seems to

Table 1: Accuracy on ImageNet linear evaluation. $x \Leftrightarrow y$ denotes standard contrastive matching between views. In "InfoNCE$_S$", we use the same base InfoMin Aug. architecture but augments the loss function with conditional MI maximization across views ($x \Leftrightarrow_{x''} y$). All models use a standard Resnet-50 architecture. ($\uparrow$) represents improvement over InfoMin Aug.

| Model | Views | Epoch | Top-1 | Top-5 |
|---|---|---|---|---|
| SimCLR | $x \Leftrightarrow y$ | 200 | 66.6 | - |
| MocoV2 | $x \Leftrightarrow y$ | 200 | 67.5 | - |
| InfoMin Aug. | $x \Leftrightarrow y$ | 200 | 70.1 | 89.5 |
| +InfoNCE$_S$ ($x'' = \texttt{cut}(x)$) | $x \Leftrightarrow y, x'' \Leftrightarrow y, x \Leftrightarrow_{x''} y$ | 200 | 70.6 ($\uparrow$) | 89.8 ($\uparrow$) |
| +InfoNCE$_S$ ($x'' = \texttt{crop}(x)$) | $x \Leftrightarrow y, x'' \Leftrightarrow y, x \Leftrightarrow_{x''} y$ | 200 | 70.9 ($\uparrow$) | 90.1 ($\uparrow$) |
|   without cond. MI ($x'' = \texttt{crop}(x)$) | $x \Leftrightarrow y, x'' \Leftrightarrow y$ | 200 | 69.9 ($\downarrow$) | 89.5 (-) |
| *Additional Losses / More Epochs* | | | | |
| SwAV | - | 200 | 72.0 | - |
| ByOL | - | 800 | 74.3 | 91.6 |

estimate MI reliably. Its empirical behavior for MI = $\{5, 10\}$ could indicate that InfoNCEs IS is a valid lower bound on MI, although we couldn't prove it formally.

## 5.2 VISION

**Imagenet** We study self-supervised learning of image representations using 224x224 images from ImageNet. The evaluation is performed by fitting a linear classifier to the task labels using the pre-trained representations only, that is, we fix the weights of the pre-trained image encoder $f$. Each input image is independently augmented into two views $x$ and $y$ using a stochastically applied transformation. For the base model hyper-parameters and augmentations, we follow the "InfoMin Aug." setup (Tian et al., 2020). This uses random resized crop, color jittering, gaussian blur, rand augment, color dropping, and jigsaw as augmentations and uses a momentum-contrastive memory buffer of $K = 65536$ examples (Chen et al., 2020b).

We fork $x$ into two sub-views $\{x', x''\}$: we set $x' \triangleq x$ and $x''$ to be an information-restricted view of $x$. We found beneficial to maximize both decompositions of the MI: $I(x'; y) + I(x''; y|x') = I(x''; y) + I(x'; y|x'')$. By noting that $I(x''; y|x')$ is likely zero given that the information of $x''$ is contained in $x'$, our encoder $f$ is trained to maximize:

$$\mathcal{L} = \lambda I_{NCE}(x'; y|f, K) + (1 - \lambda) \left( I_{NCE}(x''; y|f, K) + I_{IS}(x'; y|x'', f, K) \right) \quad (10)$$

Note that if $x'' = x$, then our decomposition boils down to maximizing the standard InfoNCE bound. Therefore, InfoMin Aug. is recovered by fixing $\lambda = 1$ or by setting $x'' = x$. The computation of the conditional MI term does not add computational cost as it can be computed by caching the logits used in the two unconditional MI terms (see Sec. B).

We experiment with two ways of obtaining restricted information views $x''$: $\texttt{cut}$, which applies cutout to $x$, and $\texttt{crop}$ which is inspired by Caron et al. (2020) and consists in cropping the image aggressively and resizing the resulting crops to 96x96. To do so, we use the $\texttt{RandomResizedCrop}$ from the $\texttt{torchvision.transforms}$ module with parameters: $\texttt{s} = (0.05, 0.14)$. Results are reported in Table 1. Augmenting the InfoMin Aug. base model with our conditional contrastive loss leads to 0.8% gains on top-1 accuracy and 0.6% on top-5 accuracy. We notice that the $\texttt{crop}$ strategy seems to perform slightly better than the $\texttt{cut}$ strategy. One reason could be that cutout introduces image patches that do not follow the pixel statistics in the corpus. More generally, we think there could be information restricted views that are better suited than others. In order to isolate the impact on performance due to integrating an additional view $x''$, i.e. the $I_{NCE}(x''; y|f, K)$ term in the optimization, we set the conditional mutual information term to zero in the line "without cond. MI". We see that this does not improve over the baseline InfoMin Aug., and its performance is 1% lower than our method, pointing to the fact that maximizing conditional MI across views provides the observed gains. We also include the very recent results of SwAV (Caron et al., 2020) and ByOL (Grill et al., 2020) which use a larger number of views (SwAV) and different loss functions (SwAV, ByOL)

and thus we think are orthogonal to our approach. We think our approach is general and could be integrated in those solutions as well.

**CIFAR-10** We also experiment on CIFAR-10 building upon SimCLR (Chen et al., 2020b), which uses a standard ResNet-50 architecture by replacing the first 7x7 Conv of stride 2 with 3x3 Conv of stride 1 and also remove the max pooling operation. In order to generate the views, we use Inception crop (flip and resize to 32x32) and color distortion. We train with learning rate 0.5, batch-size 800, momentum coefficient of 0.9 and cosine annealing schedule. Our energy function is the cosine similarity between representations scaled by a temperature of 0.5 (Chen et al., 2020b). We obtain a top-1 accuracy of 94.7% using a linear classifier compared to 94.0% as reported in Chen et al. (2020b) and 95.1% for a supervised baseline with same architecture.

### 5.3 DIALOGUE

For dialogue language modeling, we adopt the predictive coding framework (Elias, 1955; McAllester & Stratos, 2018) and consider past and future in a dialogue as views of the same conversation. Given $L$ utterances $x = (x_1, \ldots, x_L)$, we maximize $I_{NCS}(x_{\leq k}; x_{>k}|f, K)$, where past $x_{\leq k} = (x_1, \ldots, x_k)$ and future $x_{>k} = (x_{k+1}, \ldots, x_L)$ are obtained by choosing a split point $1 < k < L$. We obtain $f(x_{\leq k}), f(x_{>k})$ by computing a forward pass of the fine-tuned "small" GPT2 model (Radford et al., 2019) on past and future tokens, respectively, and obtaining the state corresponding to the last token in the last layer.

We evaluate our introduced models against different baselines. GPT2 is a basic small pre-trained model fine-tuned on the dialogue corpus. TransferTransfo (Wolf et al., 2019) augments the standard next-word prediction loss in GPT2 with the next-sentence prediction loss similar to Devlin et al. (2019). Our baseline GPT2+InfoNCE maximizes $I_{NCE}(x_{\leq k}; x_{>k}|f, K)$ in addition to standard next-word prediction loss. In GPT2+InfoNCE$_S$, we further set $x' = x_{\leq k}$ and $x'' = x_k$, the recent past, and maximize $I_{NCES}(x_{\leq k}, x_{>k})$. To maximize the conditional MI bound, we sample contrastive futures from $p(x_{>k}|x_k; \theta_{\text{GPT2}})$, using GPT2 itself as the variational approximation[4].

We fine-tune all models on the Wizard of Wikipedia (WoW) dataset (Dinan et al., 2018) with early stopping on validation perplexity. We evaluate our models using automated metrics and human evaluation: we report perplexity (*ppl*), BLEU (Papineni et al., 2002), and word-repetition-based metrics from Welleck et al. (2019), specifically: *seq-rep-n* measures the portion of duplicate n-grams and *seq-rep-avg* averages over $n \in \{2, 3, 4, 5, 6\}$. We measure diversity via *dist-n* (Li et al., 2016), the number of unique $n$-grams, normalized by the total number of $n$-grams.

Table 4 shows results on the validation set. For the test set results, please refer to the Appendix. Incorporating InfoNCE yields improvements in all metrics[5]. Please refer to the Appendix for sample dialogue exchanges. We also perform human evaluation on randomly sampled 1000 WoW dialogue contexts. We present the annotators with a pair of candidate responses consisting of GPT2+InfoNCE$_S$ responses and baseline responses. They were asked to compare the pairs regarding interestingness, relevance and humanness, using a 3-point Likert scale (Zhang et al., 2019). Table 4 lists the difference between fraction of wins for GPT2+InfoNCE$_S$ and other models as *H-rel*, *H-hum*, and *H-int*. Overall, GPT2+InfoNCE$_S$ was strongly preferred over GPT2, TransferTransfo and GPT2+InfoNCE, but not the gold response. Bootstrap confidence intervals and p-values (t-test) indicate all improvements except for GPT2+InfoNCE on the relevance criterion are significant at $\alpha$=0.05.

## 6 DISCUSSION

The result in Eq. 5 is reminiscent of conditional noise-contrastive estimation (CNCE) (Ceylan & Gutmann, 2018) which proposes a framework for data-conditional noise distributions for noise contrastive estimation (Gutmann & Hyvärinen, 2012). Here, we provide an alternative interpretation in terms of a bound on conditional mutual information. In CNCE, the proposal distribution is obtained by noising the conditional proposal distribution. It would be interesting to investigate whether it

---

[4]The negative sampling of future candidates is done offline.

[5]Note that our results are not directly comparable with Li et al. (2019) as their model is trained from scratch on a not publicly available Reddit-based corpus.

Table 2: Results for perplexity, sequence-level metric, token-level metrics, BLEU, diversity metrics and human evaluation on the valid data of the Wizard of Wikipedia dataset (Dinan et al., 2018).

| Model | ppl | seq-rep | rep | wrep | uniq | dist-1 | dist-2 | BLEU | H-rel | H-hum | H-int |
|---|---|---|---|---|---|---|---|---|---|---|---|
| GPT2 | 19.21 | 0.057 | 0.133 | 0.128 | 8276 | 0.065 | 0.389 | 0.776 | 0.20 | 0.12 | 0.35 |
| TransferTransfo | 19.32 | 0.077 | 0.132 | **0.129** | 7735 | 0.064 | 0.390 | 0.754 | 0.17 | 0.09 | 0.27 |
| GPT2+InfoNCE | 18.85 | 0.054 | 0.132 | 0.131 | 8598 | 0.064 | 0.382 | 0.798 | 0.05 | 0.10 | 0.08 |
| GPT2+InfoNCE$_S$ | **18.70** | **0.053** | **0.131** | **0.129** | **8673** | **0.066** | **0.401** | **0.816** | 0 | 0 | 0 |
| Ground Truth | – | 0.052 | 0.095 | – | 9236 | 0.069 | 0.416 | – | −0.32 | −0.34 | −0.14 |

is possible to form information-restricted views by similar noise injection, and whether "optimal" info-restricted views exist.

Recent work questioned whether MI maximization itself is at the core of the recent success in representation learning (Rainforth et al., 2018; Tschannen et al., 2019). These observed that models capturing a larger amount of mutual information between views do not always lead to better downstream performance and that other desirable properties of the representation space may be responsible for the improvements (Wang & Isola, 2020). Although we acknowledge that various factors can be at play for downstream performance, we posit that devising more effective ways to maximize MI will still prove useful in representation learning, especially if paired with architectural inductive biases or explicit regularization methods.

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

# A DERIVATIONS

## A.1 DERIVATION OF INFONCE, $I_{NCE}$

We start from Barber and Agakov's variational lower bound on MI (Barber & Agakov, 2003). $I(x; y)$ can be bounded as follows:

$$I(x; y) = \mathbb{E}_{p(x,y)} \log \frac{p(y|x)}{p(y)} \geq \mathbb{E}_{p(x,y)} \log \frac{q(y|x)}{p(y)}, \tag{11}$$

where $q$ is an arbitrary distribution. We show that the InfoNCE bound (Oord et al., 2018) corresponds to a particular choice for the variational distribution $q$ followed by the application of the Jensen inequality. Specifically, $q(y|x)$ is defined by independently sampling a set of examples $\{y_1, \ldots, y_K\}$ from a proposal distribution $\pi(y)$ and then choosing $y$ from $\{y_1, \ldots, y_K\}$ in proportion to the importance weights $w_y = \frac{e^{E(x,y)}}{\sum_k e^{E(x,y_k)}}$, where $E$ is a function that takes $x$ and $y$ and outputs a scalar. In the context of representation learning, $E$ is usually a dot product between some representations of $x$ and $y$, e.g. $f(x)^T f(y)$ (Oord et al., 2018). The unnormalized density of $y$ given a specific set of samples $y_{2:K} = \{y_2, \ldots, y_K\}$ and $x$ is:

$$q(y|x, y_{2:K}) = \pi(y) \cdot \frac{K \cdot e^{E(x,y)}}{e^{E(x,y)} + \sum_{k=2}^{K} e^{E(x,y_k)}}, \tag{12}$$

where we introduce a factor $K$ which provides "normalization in expectation". By normalization in expectation, we mean that taking the expectation of $q(y|x, y_{2:K})$ with respect to resampling of the alternatives $y_{2:K}$ from $\pi(y)$ produces a normalized density (see Sec. A.1.1 for a derivation):

$$\bar{q}(y|x) = \mathbb{E}_{\pi(y_{2:K})}[q(y|x, y_{2:K})], \tag{13}$$

where $\pi(y_{2:K}) = \prod_{k=2}^{K} \pi(y_k)$. The InfoNCE bound (Oord et al., 2018) is then obtained by setting the proposal distribution as the marginal distribution, $\pi(y) \equiv p(y)$ and applying Jensen's inequality, giving:

$$I(x, y) \geq \mathbb{E}_{p(x,y)} \log \frac{\mathbb{E}_{p(y_{2:K})} q(y|x, y_{2:K})}{p(y)} \geq \mathbb{E}_{p(x,y)} \left[ \mathbb{E}_{p(y_{2:K})} \log \frac{p(y) \, K \cdot w_y}{p(y)} \right]$$

$$= \mathbb{E}_{p(x,y)} \left[ \mathbb{E}_{p(y_{2:K})} \log \frac{K \cdot e^{E(x,y)}}{e^{E(x,y)} + \sum_{k=2}^{K} e^{E(x,y_k)}} \right]$$

$$= \mathbb{E}_{p(x,y_1)p(y_{2:K})} \left[ \log \frac{e^{E(x,y)}}{\frac{1}{K} \sum_{k=1}^{K} e^{E(x,y_k)}} \right] = I_{NCE}(x; y|E, K) \leq \log K, \tag{14}$$

where the second inequality has been obtained using Jensen's inequality.

## A.1.1 DERIVATION OF NORMALIZED DISTRIBUTION

We follow Cremer et al. (2017) to show that $q(y|x) = \mathbb{E}_{y_{2:K} \sim \pi(y)}[q(y|x, y_{2:K})]$ is a normalized distribution:

$$\int_x q(y|x) \, dy = \int_y \mathbb{E}_{y_{2:K} \sim \pi(y)} \left( \pi(y) \frac{e^{E(x,y)}}{\frac{1}{K} \left( \sum_{k=2}^{K} e^{E(x,y_k)} + e^{E(x,y)} \right)} \right) dy$$

$$= \int_y \pi(y) \mathbb{E}_{y_{2:K} \sim \pi(y)} \left( \frac{e^{E(x,y)}}{\frac{1}{K} \left( \sum_{k=2}^{K} e^{E(x,y_k)} + e^{E(x,y)} \right)} \right) dy$$

$$= \mathbb{E}_{\pi(y)} \mathbb{E}_{\pi(y_{2:K})} \left( \frac{e^{E(x,y)}}{\frac{1}{K} \left( \sum_{k=2}^{K} e^{E(x,y_k)} + e^{E(x,y)} \right)} \right)$$

$$= \mathbb{E}_{\pi(y_{1:K})} \left( \frac{e^{E(x,y)}}{\frac{1}{K} \sum_{k=1}^{K} e^{E(x,y_k)}} \right)$$

$$= K \cdot \mathbb{E}_{\pi(y_{1:K})} \left( \frac{e^{E(x,y_1)}}{\sum_{k=1}^{K} e^{E(x,y_k)}} \right)$$

$$= \sum_{i=1}^{K} \mathbb{E}_{\pi(y_{1:K})} \frac{e^{E(x,y_i)}}{\sum_{k=1}^{K} e^{E(x,y_k)}}$$

$$= \mathbb{E}_{\pi(y_{1:K})} \frac{\sum_{i=1}^{K} e^{E(x,y_i)}}{\sum_{k=1}^{K} e^{E(x,y_k)}} = 1 \tag{15}$$

## A.2 Proofs for $I_{CNCE}$

**Proposition 1** (**Conditional InfoNCE**). *The following is a lower-bound on the conditional mutual information $I(x'; y|x'')$ and verifies the properties below:*

$$I_{CNCE}(x'; y|x'', E, K) = \mathbb{E}_{p(x', x'', y_1)p(y_{2:K}|x'')} \left[ \log \frac{e^{E(x'', x', y_1)}}{\frac{1}{K} \sum_{k=1}^{K} e^{E(x'', x', y_k)}} \right] \quad (6)$$

1. *$I_{CNCE} \leq I(x'; y|x'')$.*

2. *$E^* = \arg\sup_E I_{CNCE} = \log \frac{p(y|x'', x')}{p(y|x'')} + c(x', x'')$.*

3. *When $K \to \infty$ and $E = E^*$, we recover the true conditional MI:*
   *$\lim_{K \to \infty} I_{CNCE}(x'; y|x'', E^*, K) = I(x'; y|x'')$.*

*Proof.* We begin with 1., the derivation is as follows:

$$I(x'; y|x'') = \mathbb{E}_{p(x'', x', y)} \log \frac{p(y|x'', x')}{p(y|x'')} \geq \mathbb{E}_{p(x'', x', y)} \log \frac{\bar{q}(y|x'', x')}{p(y|x'')} \quad (16)$$

$$= \mathbb{E}_{p(x'', x', y)} \log \frac{\mathbb{E}_{p(y_{2:K}|x'')} q(y|x'', x', y_{2:K})}{p(y|x'')} \quad (17)$$

$$\geq \mathbb{E}_{p(x'', x', y)} \mathbb{E}_{p(y_{2:K}|x'')} \log \frac{p(y|x'') K \cdot w_y}{p(y|x'')} \quad (18)$$

$$= \mathbb{E}_{p(x'', x', y)} \mathbb{E}_{p(y_{2:K}|x'')} \log \frac{K \cdot e^{E(x'', x', y)}}{\sum_{k=1}^{K} e^{E(x'', x', y_k)}} \quad (19)$$

$$= \mathbb{E}_{p(x'', x', y)} \mathbb{E}_{p(y_{2:K}|x'')} \log \frac{e^{E(x'', x', y)}}{\frac{1}{K} \sum_{k=1}^{K} e^{E(x'', x', y_k)}} \quad (20)$$

$$= I_{CNCE}(x'; y|x'', E, K), \quad (21)$$

where we used in Eq. 16 the Jensen's inequality following Barber and Agakov's bound (Barber & Agakov, 2003) and used $p(y|x'')$ as our proposal distribution for the variational approximation $\bar{q}(y|x'', x')$.

For 2., we rewrite $I_{CNCE}$ by grouping the expectation w.r.t $x''$:

$$\mathbb{E}_{p(x'')} \left[ \mathbb{E}_{p(x', y_1|x'')p(y_{2:K}|x'')} \left[ \log \frac{e^{E(x'', x', y_1)}}{\frac{1}{K} \sum_{k=1}^{K} e^{E(x'', x', y_k)}} \right] \right]. \quad (22)$$

Given that both distributions in the inner-most expectation condition on the same $x''$, this term has the same form as $I_{NCE}$ and therefore the optimal solution is $E_{x''}^* = \log \frac{p(y|x', x'')}{p(y|x'')} + c_{x''}(x')$ (Ma & Collins, 2018). The optimal $E$ for $I_{CNCE}$ is thus obtained by choosing $E(x'', x', y) = E_x^{*''}$ for each $x''$, giving $E^* = \log \frac{p(y|x', x'')}{p(y|x'')} + c(x', x'')$.

For proving 3., we substitute the optimal critic and take the limit $K \to \infty$. We have:

$$\lim_{K \to \infty} \mathbb{E}_{p(x'', x', y_1)p(y_{2:K}|x'')} \left[ \log \frac{\frac{p(y|x'', x')}{p(y|x'')}}{\frac{1}{K} \left( \frac{p(y_1|x'', x')}{p(y_1|x'')} + \sum_{k=2}^{K} \frac{p(y_k|x'', x')}{p(y_k|x'')} \right)} \right], \quad (23)$$

From the Strong Law of Large Numbers, we know that as $\frac{1}{K-1} \sum_{k=1}^{K-1} \frac{p(y_k|x'', x')}{p(y_k|x'')} \to \mathbb{E}_{p(y|x'')} \frac{p(y|x'', x')}{p(y|x'')} = 1$, as $K \to \infty$ a.s., therefore (relabeling $y = y_1$):

$$I_{CNCE} \sim_{K \to \infty} \mathbb{E}_{p(x'', x', y)} \left[ \log \frac{\frac{p(y|x'', x')}{p(y|x'')}}{\frac{1}{K} \left( \frac{p(y|x'', x')}{p(y|x'')} + K - 1 \right)} \right] \quad (24)$$

$$\sim_{K \to \infty} \mathbb{E}_{p(x'', x', y)} \left[ \log \frac{p(y|x'', x')}{p(y|x'')} + \log \frac{K}{\left( \frac{p(y|x'', x')}{p(y|x'')} + K - 1 \right)} \right] \quad (25)$$

$$\sim_{K \to \infty} I(x', y|x''), \quad (26)$$

where the last equality is obtained by noting that the second term $\to 0$. $\square$

## A.3 Proofs for $I_{VAR}$

**Proposition 2 (Variational $I_{CNCE}$).** *For any variational approximation $\tau(y|x'')$ in lieu of $p(y|x'')$,*

$$I_{VAR}(x', y|x'', E, \tau, K) = \mathbb{E}_{p(x', x'', y_1)\tau(y_{2:K}|x'')} \left[ \log \frac{e^{E(x'', x', y_1)}}{\frac{1}{K}\sum_{k=1}^{K} e^{E(x'', x', y_k)}} \right] \tag{7}$$
$$- \mathbb{E}_{p(x'')} \left[ KL\left( p(y|x'') \| \tau(y|x'') \right) \right],$$

*with $p(\cdot|x'') << \tau(\cdot|x'')$ for any $x''$, we have the following properties:*

1. $I_{VAR} \le I(x'; y|x'')$.

2. *If $\tau(y|x'') = p(y|x'')$, $I_{VAR} = I_{CNCE}$.*

3. $\lim_{K\to\infty} \sup_E I_{VAR}(x'; y|x'', E, \tau, K) = I(x'; y|x'')$.

*Proof.* For 1., we proceed as follows:

$$I(x'; y|x'') \ge \mathbb{E}_{p(x,y)} \left[ \log \frac{q(y|x'', x')\tau(y|x'')}{p(y|x'')\tau(y|x'')} \right]$$
$$= \mathbb{E}_{p(x,y)} \left[ \log \frac{q(y|x'', x')}{\tau(y|x'')} \right] - \mathbb{E}_{p(x)} \left[ KL(p(y|x'')\|\tau(y|x'')) \right]$$
$$\ge \mathbb{E}_{p(x,y_1)\tau(y_{2:K}|x'')} \left[ \log \frac{e^{E(x'', x', y_1)}}{\frac{1}{K}\sum_{k=1}^{K} e^{E(x'', x', y_1)}} \right] - \mathbb{E}_{p(x)} \left[ KL(p(y|x'') \| \tau(y|x'')) \right],$$
$$= I_{VAR}(x', y|x'', E, \tau, K) \tag{27}$$

where the last step has been obtained as in Eq. 18.

Proving 2. is straightforward by noting that if $\tau = p$, $KL(p(y|x'')\|\tau(y|x'')) = 0$ and the first term corresponds to $I_{CNCE}$.

Proving 3. goes as follows:

$$\sup_E \mathbb{E}_{p(x', x'', y_1)\tau(y_{2:K}|x'')} \left[ \log \frac{e^{E(x'', x', y_1)}}{\frac{1}{K}\sum_{k=1}^{K} e^{E(x'', x', y_k)}} \right] - \mathbb{E}_{p(x'')} \left[ KL\left( p(y|x'') \| \tau(y|x'') \right) \right] \tag{28}$$

$$= E_{p(x'', x', y_1)\tau(y_{2:K}|x'')} \left[ \log \frac{p(y_1|x'', x')}{\tau(y_1|x'')} - \log \frac{p(y_1|x'')}{\tau(y_1|x'')} - \log \frac{1}{K}\sum_{k=1}^{K} \frac{p(y_k|x', x'')}{\tau(y_k|x'')} \right] \tag{29}$$

$$= I(x', y|x'') - \mathbb{E}_{p(x'', x', y_1)\tau(y_{2:K}|x'')} \left[ \log \frac{1}{K}\sum_{k=1}^{K} \frac{p(y_k|x', x'')}{\tau(y_k|x'')} \right] \tag{30}$$

$$\to_{K\to\infty} I(x', y|x''). \tag{31}$$

This is obtained by noting that (1) for any $K$ and $\tau$, $\arg\sup_E I_{VAR} = \frac{p(y|x'', x')}{\tau(y|x')}$ (because the KL doesn't depend on $E$) and (2) the second term in the last line goes to 0 for $K \to \infty$ (a straightforward application of the Strong Law of Large Numbers shows that for samples $y_{2:K}$ drawn from $\tau(y_{2:K}|x'')$, we have: $\frac{1}{K}\sum_{k=2}^{K} \frac{p(y_k|x', x'')}{\tau(y_k|x'')} \to_{K\to\infty} 1$). $\square$

## A.4 Proofs for $I_{IS}$

We will be using the following lemma.

**Lemma 1.** *For any $x''$, $x'$ and $y$, and any sequence $E_K$ such that $||E_K - E||_\infty \to_{K\to\infty} 0$:*

$$\lim_{K\to\infty} \mathbb{E}_{p(y_{2:K})} \log \frac{Ke^{E_K(x'', x', y)}}{e^{E_K(x'', x', y)} + (K-1)\sum_{k=2}^{K} w_k e^{E_K(x'', x', y_k)}} \tag{32}$$

$$= \lim_{K\to\infty} \mathbb{E}_{p(y_{2:K}|x'')} \log \frac{Ke^{E(x'', x', y)}}{e^{E(x'', x', y)} + \sum_{k=2}^{K} e^{E(x'', x', y_k)}}, \tag{33}$$

*where $w_k = \frac{\exp \bar{E}(x'', y_k)}{\sum_{k=2}^{K} \exp \bar{E}(x'', y_k)}$ for $\bar{E}(x'', y_k) = \arg\sup_E I_{NCE}(x'', y|E, K) = \frac{p(y_k|x'')}{p(y_k)}$.*

*Proof.* We see that almost surely, for $y_{2:K} \sim p(\cdot)$:

$$\sum_{k=2}^{K} w_k e^{E_K(x'',x',y_k)} = \frac{\frac{1}{K-1}\sum_{k=2}^{K}\frac{p(y_k|x'')}{p(y_k)}e^{E_K(x'',x',y_k)}}{\frac{1}{K-1}\sum_{k=2}^{K}\frac{p(y_k|x'')}{p(y_k)}} \to_{K\to\infty} \mathbb{E}_{p(y|x'')}e^{E(x'',x',y)}, \qquad (34)$$

where we applied the Strong Law of Large Numbers to the denominator.

For the numerator, we write:

$$\frac{1}{K-1}\sum_{k=2}^{K}\frac{p(y_k|x'')}{p(y_k)}e^{E_K(x'',x',y_k)} = \frac{1}{K-1}\sum_{k=2}^{K}\frac{p(y_k|x'')}{p(y_k)}e^{E(x'',x',y_k)}$$

$$+ \frac{1}{K-1}\sum_{k=2}^{K}\frac{p(y_k|x'')}{p(y_k)}(e^{E_K(x'',x',y_k)} - e^{E(x'',x',y_k)})$$

and note that the first term is the standard IS estimator using $p(y_k)$ as proposal distribution and tends to $\mathbb{E}_{p(y|x'')}e^{E(x'',x',y)}$ from the Strong Law of Large Numbers, while the second term goes to 0 as $E_K$ tends to $E$ uniformly.

This gives $\lim_{K\to\infty} \mathbb{E}_{p(y_{2:K})} \log \frac{Ke^{E_K(x'',x',y)}}{e^{E_K(x'',x',y)}+(K-1)\sum_{k=2}^{K}w_k e^{E_K(x'',x',y_k)}} = \log \frac{e^{E(x'',x',y)}}{\mathbb{E}_{p(y|x'')}e^{E(x'',x',y)}}$.

Following the same logic, without the importance-sampling demonstrates that:

$$\lim_{K\to\infty} \mathbb{E}_{p(y_{2:K}|x'')} \log \frac{Ke^{E(x'',x',y)}}{e^{E(x'',x',y)} + \sum_{k=2}^{K}e^{E(x'',x',y_k)}} = \log \frac{e^{E(x'',x',y)}}{\mathbb{E}_{p(y|x'')}e^{E(x'',x',y)}},$$

which concludes the proof. $\qquad\square$

**Proposition 3 (Importance Sampling $I_{CNCE}$).** *The following approximation of $I_{SIR}$:*

$$I_{IS}(x',y|x'',E,K) = \mathbb{E}_{p(x'',x',y_1)p(y_{2:K})} \log \frac{e^{E(x'',x',y_1)}}{\frac{1}{K}(e^{E(x'',x',y_1)} + (K-1)\sum_{k=2}^{K}w_k e^{E(x'',x',y_k)})}, \qquad (9)$$

*where $w_k = \frac{\exp \bar{E}(x'',y_k)}{\sum_{k=2}^{K}\exp \bar{E}(x'',y_k)}$ and $\bar{E} = \arg\sup_E I_{NCE}(x'',y|E,K)$, verifies:*

1. $\lim_{K\to\infty} \sup_E I_{IS}(x';y|x'',E,K) = I(x';y|x'')$,

2. $\lim_{K\to\infty} \arg\sup_E I_{IS} = \log \frac{p(y|x'',x')}{p(y|x'')} + c(x',x'')$.

*Proof.* By applying Lemma 1 with $E_K = E$, we know that for any $E$:

$$\lim_{K\to\infty} I_{IS}(x';y|x'',E,\bar{E},K) = \lim_{K\to\infty} \mathbb{E}_{p(x'',x',y)p(y_{2:K}|x'')} \log \frac{Ke^{E(x'',x',y)}}{e^{E(x'',x',y)} + \sum_{k=2}^{K}e^{E(x'',x',y_k)}}.$$

In particular, the RHS of the equality corresponds to $\lim_{K\to\infty} I_{CNCE}(x',y|x'',E,K)$. That quantity is smaller than $I(x',y|x'')$, with equality for $E = E^*$. This guarantees that:

$$\lim_{K\to\infty}\sup_E I_{IS}(x';y|x'',E,\bar{E},K) \geq \lim_{K\to\infty} I_{IS}(x';y|x'',E^*,\bar{E},K) = I(x',y|x''). \qquad (35)$$

We now prove the reverse inequality. We let $2\epsilon = \lim_{K\to\infty}\sup_E I_{IS}(x';y|x'',E,\bar{E},K) - I(x',y|x'')$, and assume toward a contradiction that $\epsilon > 0$. We know that:

$$\exists K_0, \quad \forall K \geq K_0, \quad \sup_E I_{IS}(x';y|x'',E,\bar{E},K) \geq I(x',y|x'') + \epsilon.$$

Now, $\forall K \geq K_0$, let $E_K$ be such that:

$$I_{IS}(x';y|x'',E_K,\bar{E},K) \geq \sup_E I_{IS}(x';y|x'',E,\bar{E},K) - \frac{\epsilon}{2},$$

and thus: $\forall K \geq K_0, I_{IS}(x';y|x'',E_K,\bar{E},K) \geq I(x',y|x'') + \frac{\epsilon}{2}$.

Since $E_K \in \mathbb{R}^{|\mathcal{X}|\times|\mathcal{X}|\times|\mathcal{Y}|}$, $\{E_K\}_{K\geq K_0}$ contains a subsequence that converges to a certain $E_\infty \in \bar{\mathbb{R}}^{|\mathcal{X}|\times|\mathcal{X}|\times|\mathcal{Y}|}$. Without loss of generality, we assume that $\forall K, \forall x'', \forall x', \mathbb{E}_{p(y)}[E_K(x'',x',y)] = 0$ which implies that $\mathbb{E}_{p(y)}[E_\infty(x'',x',y)] = 0$ (similarly to $I_{NCE}$, $I_{IS}$ is invariant to constants added to $E$).

In particular, this guarantees that $||E_\infty||_\infty < \infty$. Otherwise, we would have $E_\infty(x'',x',y) = -\infty$ for a given $y$, which would then imply $I_{IS}(x';y|x'',E_\infty,\bar{E},K) = -\infty$ and give a contradiction.

We can now apply Lemma 1 to $\{E_K\}$ and $E_\infty$ to show that $\lim_{K\to\infty} I_{IS}(x';y|x'',E_K,\bar{E},K) = \lim_{K\to\infty} I_{CNCE}(x',y|x'',E_\infty,K)$, and get a contradiction: the first term is larger than $I(x',y|x'') + \frac{\epsilon}{2}$ while the second is smaller than $I(x',y|x'')$. $\qquad\square$

# B  PSEUDOCODE

## B.1  LOSS COMPUTATION

We provide a pseudo-code for the loss computation which uses MocoV2 backbone comprising a memory of contrastive examples obtained using a momentum-averaged encoder (Chen et al., 2020b).

```
def compute_loss(xp, xpp, y, f, f_ema, memory, lam=0.5):
    """
    Args:
        xpp: info-restricted view
        xp: a view
        y: a view
        f: standard encoder
        f_ema: momentum averaged encoder
        memory: memory bank of representations

    Returns:
        lam * mi(xp; y) + (1 - lam) * (mi(xpp; y) + mi(xp; y | xpp))
    """
    # encode xp and xpp with standard encoder, (1, dim)
    q_xp, q_xpp = f(x_p), f(x_pp)
    # encode y with momentum-averaged encoder, (1, dim)
    k_y = f_ema(y).detach()
    # (1 + n_mem,), first is xpp_y score
    logits_xpp_y = dot(q_xpp, cat(k_y, memory))
    # (1 + n_mem,), first is xp_y score
    logits_xp_y = dot(q_xp, cat(k_y, memory))
    # infonce bound between xp and y
    nce_xp_y = -log_softmax(logits_xp_y)[0]
    # infonce bound between xpp and y
    nce_xpp_y = -log_softmax(logits_xpp_y)[0]
    K = len(logits_xpp_y)
    # compute resampling importance weights
    w_pp_y = softmax(logits_xpp_y[1:])
    # form approximation to the partition function (Eq. 12)
    Z_xp_y = (K - 1) * w_pp_y * exp(logits_xp_y[1:])
    Z_xp_y = Z_xp_y.sum() + exp(logits_xp_y[0])
    # infonce bound on the conditional mutual information
    nce_xp_y_I_xpp = -logits_xp_y[0] + log(Z_xp_y)
    # compose final loss
    loss = lam * nce_xp_y
    loss += (1-lam) * (nce_xpp_y + nce_xp_y_I_xpp)
    return loss
```

## B.2  SYNTHETIC EXPERIMENTS

Here, we provide details for Sec. 5.1. In this experiment, each $x'$, $x''$ and $y$ are 20-dimensional. For each dimension, we sampled $(x_i', x_i'', y_i)$ from a correlated Gaussian with mean 0 and covariance matrix $\texttt{cov}_i$. For a given value of MI, $\texttt{mi} = \{5, 10, 15, 20\}$, we sample covariance matrices $\texttt{cov}_i = \texttt{sample\_cov}(\texttt{mi}_i)$, such that $\sum_i \texttt{mi}_i = \texttt{mi}$, $\texttt{mi}_i$ chosen at random. We optimize the bounds by stochastic gradient descent (Adam, learning rate $5 \cdot 10^{-4}$). All encoders $f$ are multi-layer perceptrons with a single hidden layer and ReLU activation. Both hidden and output layer have size 100.

InfoNCE computes:

$$\mathbb{E}_p \left[ \log \frac{e^{f([x',x''])^T f(y)}}{e^{f([x',x''])^T f(y)} + \sum_{k=2}^{K} e^{f([x',x''])^T f(y_k)}} \right] + \log K, \quad y_{2:K} \sim p(y),$$

where the proposal is the marginal distribution $p(y)$, $E$ is chosen to be a dot product between representations, $\mathbb{E}_p$ denotes expectation w.r.t. the known joint distribution $p(x', x'', y)$ and is approximated with Monte-Carlo, $[x', x'']$ denotes concatenation and $f$ is a 1-hidden layer MLP.

InfoNCEs computes:

$$\mathbb{E}_{p(x'',x',y)p(y_{2:K})} \left[ \log \frac{e^{f(x'')^T f(y)}}{e^{f(x'')^T f(y)} + \sum_{k=2}^{K} e^{f(x'')^T f(y_k)}} \right] + \tag{36}$$

$$\mathbb{E}_{p(x'',x',y)p(y_{2:K}|x'')} \left[ \log \frac{e^{f([x'',x'])^T f(y)}}{e^{f([x'',x'])^T f(y)} + \sum_{k=2}^{K} e^{f([x'',x'])^T f(y_k)}} \right] + 2 \log K$$

where $f(x)$ is just $f([x, \mathbf{0}])$ in order to re-use MLP parameters for the two terms. The negative samples of the conditional MI term come from the conditional distribution $p(y|x'')$, which is assumed to be known in this controlled setting. We maximize both lower bounds with respect to the encoder $f$.

We report pseudo-code for `sample_cov`, used to generate $3{\times}3$ covariance matrices for a fixed `mi` $= I(\{x', x''\}; y)$ and uniformly sampled $\alpha = I(x''; y)/I(\{x', x''\}; y)$:

```
def sample_cov(mi):
  alpha = random.uniform(0.1, 0.9)
  params = random.normal(0, I₆)
  # use black box optimizer (Nealder-Mead) to determine opt_params
  opt_param = arg minₓ residual(params, mi, α)
  return project_posdef(opt_params)

def project_posdef(x):
  # project x ∈ ℝ⁶ to a positive definite 3x3 matrix
  cov = zeros(3, 3)
  cov[tril_indices(3)] = x
  cov /= column_norm(cov)
  return dot(cov, cov.T)

def analytical_mi(cov):
  # compute analytical MI of 3 covariate Gaussian variables
  cov_01 = cov[:2, :2]
  cov_2 = cov[2:3, 2:3]
  mi_xp_xpp_y = 0.5 * (log(det(cov_01)) + log(det(cov_2)) - log(det(cov)))
  cov_1 = cov[1:2, 1:2]
  cov_23 = cov[1:, 1:]
  mi_xp_y = 0.5 * (log(det(cov_1)) + log(det(cov_2)) - log(det(cov_23)))
  return mi_xp_xpp_y, mi_xp_y

def residual(x, mi, α):
  # penalize difference between analytical mi and target mi, αmi
  cov = project_posdef(x)
  mi_xp_y, mi_xp_y = analytical_mi(cov)
  return (mi_xp_xpp_y - mi) ** 2 + (mi_xp_y - α * mi) ** 2
```

# C  EXPERIMENTS ON DIALOGUE

## C.1  INFONCE DETAILS

For all InfoNCE terms, given the past, the model is trained to pick the ground-truth future among a set of $N$ future candidates. This candidate set includes the ground-truth future and $N - 1$ negative futures drawn from different proposal distributions. To compute InfoNCE($f(x_{\le k}); f(x_{>k})$), we consider the ground truth future of each sample as a negative candidate for the other samples in the batch. Using this approach, the number of candidates $N$ is equated to the batch size. This ensures that negative samples are sampled from the marginal distribution $p(x_{>k})$. To compute the conditional information bound InfoNCE$_S$, we sample negative futures $p(y|x_k)$ by leveraging the GPT2 model itself, by conditioning the model only on the most recent utterance $x_k$ in the past.

## C.2  EXPERIMENTAL SETUP

Given memory constraints, the proposed models are trained with a batch size of 5 per GPU over 10 epochs, considering up to three utterances for the future and five utterances in the past. All the models are trained on 2 NVIDIA V100s. The models early-stop in the 4th epoch. We use the Adam optimizer with a learning rate of $6.25 \times 10^{-5}$, which we linearly decay to zero during training. Dropout is set to 10% on all layers. InfoNCE/InfoNCE$_S$ terms are weighted with a factor 0.1 in the loss function.

Table 3: A sample dialogue between speaker $A$ and speaker $B$ from the Wizard of Wikipedia dataset. The four rows from top to bottom are: (1) the "past" dialogue up to utterance $k$ (2) the ground-truth utterance for the next turn $k+1$ (3) generations for the next turn sampled from the "restricted context" conditional "future" distribution $p(y|x_k)$ (4) future candidates sampled from the groundtruth "future" distribution. We can see that $p(y|x_k)$ is semantically close but incoherent w.r.t to the dialogue history as it was conditioned solely on the immediate past utterance $x_k$. However, we can notice that $p(y)$ is semantically distant from $x$ as it was sampled randomly from the data distribution. The highlighted text in green correspond to the topic of the conversation. Speaker $B$ indicates that it has never done either parachuting or skydiving. $p(y|x_k)$ corresponds to the set of **hard** negatives that are closely related to the conversation. $B_1$ corresponds to the utterance generated based on the restricted context $x_k$. The utterance is on-topic but completely contradictory to what speaker $B$ has said in the past. On the other hand $B_1'$ is randomly sampled from other dialogues. We can observe that the utterance is clearly irrelevant to the conversation. Therefore, it is easier to the model to discriminate between $B_1'$ and $B_{gt}$.

| | | |
|---|---|---|
| $x$ | $A$: | I like parachuting or skydiving . |
| | $\boldsymbol{B}$: | I've never done either but they sound terrifying, not a fan of heights. |
| | $A$: | But it is interesting game. This first parachute jump in history was made by Andre Jacques. |
| | $\boldsymbol{B}$: | Oh really ? Sounds like a french name, what year did he do it ? |
| | $A$: | It done in October 22 1797. They tested his contraption by leaping from a hydrogen balloon. |
| | $\boldsymbol{B}$: | Was he successful or did he kick the bucket off that stunt? |
| | $A$: | I think its a success. The military developed parachuting tech. |
| $y \sim p(y\|x_k)$ | $B_{gt}$ | Yeah nowadays they are a lot more stable and well made. |
| $y_{1:N} \sim p(y\|x_k)$ | $B_1$: | That is great. I've been skydiving for days now . How is it ? |
| | $B_2$: | Oh I have never flown but I'm glad to know. |
| | $B_3$: | I've been dying for it since I was a kid. |
| | $B_4$: | Yes, that is why NASA had an advanced mechanics tech for months. |
| | $B_5$: | I went parachuting last Sunday and enjoyed it. |
| $y_{1:N} \sim p(y)$ | $B_1'$: | I think science fiction is an amazing genre for anything |
| | $B_2'$: | Can you imagine the world without internet access ? |
| | $B_3'$: | I am just finishing my university course and I will be a qualified pharmacist. |
| | $B_4'$: | I don't know how to be romantic. I have trouble expressing emotional attraction. |
| | $B_5'$: | I think Krav Maga is a martial art sport. That 's the reason I picked it . |

Table 4: Results for perplexity, sequence-level metric, token-level metrics, BLEU and diversity metrics on the test data of the Wizard of Wikipedia dataset. Results demonstrate that the proposed InfoNCE and InfoNCE$_S$ bounds archive lower perplexity, reduce next-token repetition and increase the number of unique next-tokens compared to the baselines GPT2 and TransferTransfo.

| Model | ppl | seq-rep | rep | wrep | uniq | dist-1 | dist-2 | BLEU |
|---|---|---|---|---|---|---|---|---|
| GPT2 | 19.24 | 0.064 | 0.130 | 0.132 | 7393 | 0.064 | 0.392 | 0.775 |
| TransferTransfo | 19.33 | 0.078 | 0.134 | 0.132 | 7735 | 0.058 | 0.386 | 0.752 |
| GPT2+InfoNCE (ours) | 18.88 | 0.065 | 0.126 | 0.131 | 8432 | 0.065 | 0.390 | 0.799 |
| GPT2+InfoNCE$_S$ (ours) | **18.76** | **0.050** | **0.120** | **0.128** | **8666** | **0.070** | **0.405** | **0.810** |
| Ground Truth | – | 0.052 | 0.095 | – | 9236 | 0.069 | 0.416 | – |

Table 5: Selected responses to the same context from different methods fine-tuned on the Wizard of Wikipedia datasets.

| Context | Generated Responses |
|---|---|
| **A:** I recently took up archery and I think I love it . Do you know anything about Archery ?
**B:** Archery is a sport / skill of using a bow to propel arrows.
**A:** And a great sport it is . Do you know where archery originated from ? | **GPT2+InfoNCE$_S$**: Archery originated from ancient Mesopotamia, and is actually practiced internationally in over 200 countries.
**GPT2+InfoNCE**: I am not sure but many traditions and principles arose during the Mesolithic and Paleolithic era.
**TransferTransfo**: Yep , you just use it for skill and using it to shoot arrows.
**GPT2**: I don't know, but I know that the old French called it archer's art. |
| **A:** What is your motivation?
**B:** That's a tough question. I 'd say my own progress. I like seeing how much I improve with something. You ?
**A:** I am retired now. Are you familiar with the work of Mehr and Meyer, well known psychologists ?
**B:** They said that " Motivation is a word that is part of the popular culture as few other psychological concepts are . " Do you agree with what they said ? | **GPT2+InfoNCE$_S$**: Yes, definitely! You should check out Mark Twain's famous motivational book ! There are theories that point out how important it is to meditate.
**GPT2+InfoNCE**: They said that in psychology research, theories and evidence, participants who enjoyed continuous experience could be more likely to be successful.
**TransferTransfo**: Absolutely, I disagree. Are you an author or a commentator?
**GPT2**: That's awesome. So , what types of differences exist between "good " motivation and bad? |

## C.3 HUMAN EVALUATION

We closely follow the protocol used in Zhang et al. (2019). Systems were paired and each response pair was presented to 3 judges in random order on a 3 point Likert scale. We use a majority vote for each response pair to decide whether system1, system2, or neither, performed better. We then bootstrap the set of majority votes to obtain a 95% confidence interval on the expected difference between system1 and system2. If this confidence interval contains 0, the difference is deemed insignificant. We also compute p-values from the confidence intervals[6].

In the following tables, "pivot" is always the system given by our full InfoNCE$_S$ model. Pairings where the pairwise confidence interval is marked with "*" have a significant difference between systems.

Human Evaluation: Which response is more *relevant*?

| cmp_sys | pivot_wins | pivot_CI | cmpsys_wins | cmpsys_CI | pairwise_CI | p |
|---|---|---|---|---|---|---|
| GPT2 | 0.48726 | (0.46, 0.52] | 0.28662 | (0.26, 0.32] | (0.15, 0.26]* | 1.24835e-12 |
| GPT2MMI | 0.65833 | (0.62, 0.7] | 0.16250 | (0.13, 0.2] | (0.43, 0.56]* | 6.11888e-42 |
| GPT2_NSP | 0.46888 | (0.44, 0.5] | 0.30043 | (0.27, 0.33] | (0.11, 0.22]* | 6.67922e-09 |
| InfoNCE | 0.41711 | (0.39, 0.45] | 0.36748 | (0.34, 0.4] | (-0.01, 0.11] | 8.09387e-02 |
| gold_response | 0.22679 | (0.2, 0.25] | 0.54325 | (0.51, 0.58] | (-0.37, -0.27]* | 3.26963e-23 |

---

[6]https://www.bmj.com/content/343/bmj.d2304

Human Evaluation: Which response is more *humanlike*?

| cmp_sys | pivot_wins | pivot_CI | cmpsys_wins | cmpsys_CI | pairwise_CI | p |
|---|---|---|---|---|---|---|
| GPT2 | 0.45084 | (0.42, 0.48] | 0.32636 | (0.3, 0.36] | (0.07, 0.18]* | 1.17277e-05 |
| GPT2MMI | 0.61734 | (0.57, 0.66] | 0.18393 | (0.15, 0.22] | (0.36, 0.5]* | 1.73160e-30 |
| GPT2_NSP | 0.43617 | (0.41, 0.47] | 0.35000 | (0.32, 0.38] | (0.03, 0.14]* | 2.92302e-03 |
| InfoNCE | 0.44630 | (0.42, 0.48] | 0.34515 | (0.32, 0.38] | (0.04, 0.16]* | 4.45383e-04 |
| gold_response | 0.22164 | (0.2, 0.25] | 0.56608 | (0.53, 0.6] | (-0.4, -0.29]* | 9.29316e-28 |

Human Evaluation: Which response is more *interesting*?

| cmp_sys | pivot_wins | pivot_CI | cmpsys_wins | cmpsys_CI | pairwise_CI | p |
|---|---|---|---|---|---|---|
| GPT2 | 0.56157 | (0.53, 0.59] | 0.21444 | (0.19, 0.24] | (0.3, 0.4]* | 2.13032e-36 |
| GPT2MMI | 0.68750 | (0.65, 0.73] | 0.12292 | (0.09, 0.15] | (0.5, 0.63]* | 6.66687e-63 |
| GPT2_NSP | 0.51931 | (0.49, 0.55] | 0.24571 | (0.22, 0.27] | (0.22, 0.33]* | 2.30585e-22 |
| InfoNCE | 0.41288 | (0.38, 0.44] | 0.33580 | (0.31, 0.37] | (0.02, 0.13]* | 5.84741e-03 |
| gold_response | 0.32384 | (0.29, 0.35] | 0.46624 | (0.44, 0.5] | (-0.2, -0.09]* | 1.08781e-03 |

