# OpenReview forum: "Decomposing Mutual Information for Representation Learning"
_ICLR.cc/2021/Conference — Reject_

### Official Review · AnonReviewer4 · 2020-10-21
**Unconvinced by approximations and experiments**

**Rating:** 5
**Confidence:** 5

**Review:**

This paper proposes a contrastive learning approach where one of the views, x, is converted into two subviews, x' and x'', and then separate InfoNCE style bounds constructed for each of I(x'';y) and I(x';y|x'') before being combined to form an overall training objective.  Critically, the second of these is based on the conditional MI, I(x';y|x''), distinguishing it from previous work using multiple views that just take the marginal I(x';y).  Estimating this conditional MI transpires to be somewhat trickier due to the additional intractability from p(y|x''), with approximations suggested to get around this.  Experiments are performed on both vision and NLP problems.


*Review Overview*


I believe the premise of the paper is quite sensible and the general idea potentially useful, albeit somewhat incremental.  However, I am rather unconvinced by the approximations used to try and estimate p(y|x''); this rather lets the paper down as I feel this is the main technical challenge with the other contributions being rather straightforward.  Further, I feel the experimental results are somewhat weak, both in terms of the fact that the gains shown are quite small, but also because there are doubts as to whether these gains are actually statistically significant at all or just a somewhat "engineered" win that does not accurately reflect the underlying approaches.


*Strengths*

+ As far as I'm aware, the suggested approach has not appeared in the literature before.
+ This is a highly active research area that is very much of interest to the ICLR community.
+ The high-level idea of using subviews and breaking down the MI in this way seems sensible and potentially useful.
+ The toy experiment in Figure 2 is helpful in showing that the approach can be helpful if p(y|x'') can be estimated well.
+ It is good to see experiments on both vision and NLP problems.
+ The paper is mostly quite easy to follow.


*Weaknesses (in roughly decreasing order of importance)*
- The experiments do not follow procedures that are clearly statistically sound and the gains could easily just be a result of luck and /or tweaking the approach until it works.  In particular, no error bars are provided for any of the results; different approaches are run for different problems without justification such that it feels like things are somewhat cherry-picked; additional heuristics are added to what is actually run without clear a priori motivation such that they feel rather based on trial and error and undermines whether the gains are actually from the suggested approach or just engineering the exact setup until it gives an improvement; some important details seem to be missing such as how lambda is chosen.
- The experimental gains are rather modest.
- The approximation methods for p(y|x'') are rather unconvincing: the variational bound requires a very complex learning problem in itself to encapsulate p(y|x'') while permitting optima that do not produce approximations that are close to this target at all, while the importance sampling approach is very much just a heuristic with rather flimsy justification.  More explanation on this is given later.
- Much of the technical content, other than how one approximates p(y|x''), is very straightforward and is not enough to warrant publication on its own.  For example, the first time anything new is introduced is not until section 3.1 at the bottom of page 4, while the conditional InfoNCE bound/proposition 1 is trivial and arguably not even new to the literature (for example, equivalent bounds are already employed in conditional MI estimation in a Bayesian experimental design context, see e.g. http://proceedings.mlr.press/v108/foster20a/foster20a.pdf).
- Though not bad, the writing of the paper could definitely be improved.  In particular, the abstract and introduction could be much easier to follow and have a clearer narrative, while the initial explanations of what x, y, and the sub-views x' and x'' represent could be much clearer.
- [Minor] I'm not a fan of the title.  "Decomposing MI" suggests you are making manipulations to the standard MI to ease estimation or focus on different things, rather than introducing additional views and then introducing separate bounds for each (the approach isn't even a true decomposition as its a bound on the original MI, not an equality).


*Approximations*


As previously mentioned, one of my biggest misgivings is with the approximations employed to try and draw samples from p(y|x'').  This is at the core of the problem and arguably why most papers have steered clear of dealing with conditional MI directly.

Starting with the variational approach, I have two issues:
- Estimating p(y|x'') in this way is extremely difficult and is almost like learning a decoder in that we have to learn how to generate full data examples.  Though this is far from fatal in the context of the overall approach, it feels like if we are able to learn a good approximation for p(y|x''), we have already solved much of the challenge contrastive learning was trying to address in the first place.
- More immediately worryingly, the resulting objective introduced (Eq 10) does not appear to guarantee that an effective contrastive learning scheme will be learned as there are unwanted ways this objective can be optimized.

To be more precise about the latter point, the paper claims about Eq 10

"This suggests that using to sample contrastive sets retains the properties of being a lower bound on the conditional MI."

This is a strange phrase and not properly quantified: my interpretation of Eq (10) would be that the KL term being positive means using a variational approximation does not produce a valid lower bound in any useful sense.  In particular, because \tau appears in both terms, it seems perfectly possible for there to be optima of Eq (10) that do not correspond to \tau(y|x'') = p(y|x'') and consequently critics E that do not behave in the desired manner.  Essentially, it seems like the approach, at least in principle, can learn a tau that allows for easy discrimination to the contrastive samples rather than which represents the actual distribution; the first term could potentially just dominate the KL divergence (such a phenomenon is well documented in VAEs) and avoid learning anything useful.

This is not the end of the world if it does actually train properly in practice, but I think this needs to be directly demonstrated and the potential pitfalls that are there discussed.

On the other hand, with the "importance sampling" approach, there are many large and not fully justified jumps being made such that the final objective used (i.e. Eq 11) is more of an importance sampling inspired heuristic than any principled importance sampling estimate.  There are a lot of approximations going on here, from the weights themselves (they are only valid weights if E=E* and its not clear they will give anything particularly sensible when E\neq E*)  to the rather unjustified jump from these to Eq 11.  In particular, deriving Eq 11 from the weights somewhat boils down to making the assumption E[1/X] = 1/E[X] which is typically a poor approximation (though interestingly might become exact in the limit K->\inf). In short, I would suggest that the approach Eq (11) requires far more careful justification for it to be convincing that this is an approach one would a priori expect to behave in the intended manner.  Some sort of ablation study into how reasonable the approximation it provides is would also be extremely helpful (this could also be done for the variational approach).


*Other Specific Comments *


1. It would be good to explain what you mean by "hard" contrastive samples in the intro.

2. The sentence

"We note that the bound (Oord et al., 2018) corresponds InfoNCE to a particular choice for the variational distribution q."

is not quite true: the InfoNCE bound can be viewed as a lower bound on Eq (2) (not Eq (2) itself) for a particular choice of q as it still requires an extra invocation of Jensen's inequality.


*Questions etc *


a. Is a stop gradient is used with the w_k to stop E from being indirectly trained through these?

b. I would like to see the results of using the variational approach with the vision problem and the importance sampling approach with the NLP problem, or at the very least a justification for why this hasn't been done.  Without this the results feel a little cherry-picked.

c. What is the performance you get when setting lambda=0?  This was the originally proposed method and, presumably, experiments were run on this so it is important to include them in the results, both so we can see how important the extra heuristic it conveys actually is, and because it relates to the statistical interpretation of the results (tuning lambda reduces the significance of the results as its an extra degree of freedom the baselines don't have).  Moreover, how was lambda actually chosen?  This seems to be missing from the paper.

d. The experimental results are pretty meaningless without any sort of error bars.  The gains are quite small and it impossible to know at present whether they are the result of pure chance, or actual statistically significant improvements.  I would like to see repeat runs and the variability in the performance across these.

e. Are the same adjustments used for the dialogue experiment as for the vision experiment?  In general, section 4.3 is quite hard to follow and seems to lack complete details for the exact setup used for the suggested approach (section 4.2 still has a few things missing as well like setting lambda).

f. What is done with the bootstrap confidence intervals?  This was difficult to understand from the shortness of the explanation, while it is rather unsatisfactory to just quote that the results were significant at the 5% level rather than actually providing the variations, p values, and exact details of the test being done.

---

> ### Author Response · Authors · 2020-11-18
> **Strengthened theory and clarifications about experiments, 1/2**
>
> Thanks for the thorough and insightful review and for the time spent in deeply understanding our paper. It was very informative to read your comments and thoughts about the paper. We provided stronger guarantees for our bounds and approximations and we added an empirical evaluation of the importance sampling estimation of conditional MI into the synthetic experiments. Please, note the answers to your (a-f) questions in the 2/2 part of this answer, specifically to your doubts on the experimental setting.
>
> **General clarity and contribution**
>
> _"Much of the technical content, other than how one approximates p(y|x''), is very straightforward and is not enough to warrant publication on its own. For example, the first time anything new is introduced is not until section 3.1 at the bottom of page 4, while the conditional InfoNCE bound/proposition 1 is trivial and arguably not even new to the literature( see http://proceedings.mlr.press/v108/foster20a/foster20a.pdf)."_
>
> We weren’t aware of Foster (2020), thanks, we have added the reference to our revision. That very recent and interesting paper proposes I_ACE in the context of BOED. We agree that in light of that paper, the conditional InfoNCE (I_CNCE) bound is similar (although we prove the form for the optimal E). Apart from I_CNCE and the proposed bounds and approximations on the conditional MI, we also feel that the idea of splitting MI into a sum of smaller MIs and maximizing each in turn (I_NCES) is novel and one of the central contributions of the paper. We stressed this point in the updated Section 3.
>
> _"Though not bad, the writing of the paper could definitely be improved. In particular, the abstract and introduction could be much easier to follow and have a clearer narrative, while the initial explanations of what x, y, and the sub-views x' and x'' represent could be much clearer"_
>
> We are super happy to improve the writing. How would you suggest we could structure the narrative?
>
> _""Decomposing MI" suggests you are making manipulations to the standard MI to ease estimation or focus on different things, rather than introducing additional views and then introducing separate bounds for each."_
>
> We feel our results are general and can be used outside of the concept of “views”. For example, for any variable x = (x_1, x_2), and y we maximize the MI by decomposing it as I(x_1, y) + I(x_2, y | x_1) etc.. This is an equality due to chain rule, and we propose to maximize each of the term in the decomposition, and we argue that we can have a tighter estimate to the total MI. Admittedly, that maximization does apply to lower-bounds of the quantities.
>
> **About the Variational Bound (I_VAR) and Importance Sampling (I_IS)**
>
> _"estimating p(y|x'') in this way is extremely difficult and is almost like learning a decoder in that we have to learn how to generate full data examples. Though this is far from fatal in the context of the overall approach, it feels like if we are able to learn a good approximation for p(y|x''), we have already solved much of the challenge contrastive learning was trying to address in the first place"_
>
> We agree that estimating p(y|x’’) is costly but could be feasible (it would amount to train a Flow decoder for images, or autoregressive decoder for text, something we do in our dialogue experiments). An imperfect \tau(y|x’’) could still be used to train more informed representations of the input datum x by maximizing conditional MI.
>
> _"More immediately worryingly, the resulting objective introduced (Eq 10) does not appear to guarantee that an effective contrastive learning scheme will be learned as there are unwanted ways this objective can be optimized."_
>
> We have added a discussion of I_VAR highlighting your connection to VAEs, as well as demonstrated some of its properties. If we don’t optimize the expectation term wrt \tau (which is how our experiments were performed), then we are guaranteed to have the same optimal solution for I_CNCE, i.e. \tau^* = p and E* = log p(y|x’’, x’)/p(y|x’’).
>
> _"I would suggest that the approach Eq (11) requires far more careful justification for it to be convincing"_
>
> We have expanded our results for I_IS showing that in the limit of K->inf, the sup_E converges to I(x’, y | x’’). We also show that the value is obtained for the critic learnt by I_CNCE but without assuming knowledge of p(y | x’’). In the synthetic experiment, we benchmarked the amount of MI estimated by I_IS, and see that is competitive with the I_CNCE, which assumes access to p(y|x’’).

---

> > ### Author Response · Authors · 2020-11-18
> > **Strengthened theory and clarifications about experiments, 2/2**
> >
> > Answers to the explicit questions below:
> >
> > a) **About w_k**: We tried with and without stopping the gradient, the results were not significantly affected by this choice. Our theorem assumes the w_k are optimal, which would correspond to detaching the weights.
> >
> > b) **About bounds used in each task**: We didn’t cherry pick how we maximize conditional MI for each task. Our justification for using the I_IS in the image setting and I_VAR in the dialogue setting is the following: as you pointed out, in the image setting using a variational q(y | x’’) would correspond to training an image decoder which could be feasible but is definitely expensive (one would have to sample K negative samples from a flow model for each x’’ in the batch…). Given that InfoMin Aug. uses a large contrastive buffer of negative samples (65536), I_IS is definitely more suited in this setting and very cheap to implement. On the contrary, in the dialogue setting, encoding a lot of negative examples with GPT-2 is very expensive (as it requires a lot of forward passes through GPT-2 which is infeasible given our computational resources), while sampling examples from q(y | x’’) is cheap, due to the fact that we can use ancestral sampling to generate possible future continuations y given a restricted past x’’ in parallel and that sentences are in general short in the context of dialogue. If you agree, we’ll write this justification in the paper, we’re sorry we missed that explanation.
> >
> > c) **About lambda=0**: The result when setting lambda=0 is lower than the InfoMin Aug. baseline and thus than our model. We’ll rerun this experiment and add the exact number to the final version of the paper. Our hypothesis is that this is due to a failure in sharing learning signals between the critics of conditional and unconditional MI. By maximizing I_NCE(x’’, y) + I_IS(x’, y | x’’) w.r.t. the same encoder f, the representation z’’ = f(x’’) is trained with negative examples from the marginal, while z’ = f(x’) is trained with negative examples from p(y | x’’). But at linear evaluation time, the model is evaluated only using z’ = f(x’). Therefore, if the encoder is large enough, x’ and x’’ are easily distinguishable by f and thus f can learn strikingly different feature spaces: z’ could contain only “global” info useful to distinguish the image from samples p(y | x’’), while z’’ could contain only “local” info useful to distinguish from samples p(y). By also maximizing the term I_NCE(x’, y), we encourage f(x’) to include also features useful to distinguish x’ from negative examples p(y) in the corpus thus bringing z’ = f(x’) and z’’ = f(x’’) distributionally closer. We are happy to add this discussion to the paper if you feel it’s relevant.
> >
> > d) **About error bars**: Each of the vision experiments take 6 days to run on our gpus. We agree that error bars would be better, but we followed the standard self-supervised representation learning evaluation where error bars are not provided (see e.g. https://arxiv.org/abs/1906.05849, https://arxiv.org/pdf/2007.09852.pdf ). We are willing to provide those but we would need more time to do so as we have to do it also for the baselines reported.
> >
> > e) **About lambda**: Yes, sorry, we followed the same experimental setting. Lambda is chosen on the validation set. We will add this to the final version.
> >
> > f) **About bootstrap**: We closely follow the protocol used in Zhang et al. 2019. Systems were paired and each response pair was presented to 3 judges in random order on a 3 point Likert scale. We use a majority vote for each response pair to decide whether system1, system2, or neither, performed better. We then bootstrap the set of majority votes to obtain a 95% confidence interval on the expected difference between system1 and system2. If this confidence interval contains 0, the difference is deemed insignificant. We also computed p-values via https://www.bmj.com/content/343/bmj.d2304. All confidence intervals and p-values are now reported in the appendix.

---

> > > ### Comment · AnonReviewer4 · 2020-11-19
> > > **Moving in right direction, but too many outstanding concerns to back acceptance**
> > >
> > > Thank you for your response and extensive paper update.  It is great to see that you have made substantial efforts to improve the paper and it is very clear that it is moving in the right direction; I have increased my score to account for this.  Unfortunately, I still do not feel I can advocate for acceptance though as:
> > >
> > > a) There are still a large number of issues outstanding with the experiments.  I appreciate that a lot of this is from a lack of time and scale of what has been asked for rather than a lack of effort to update them, but I think some aspects are quite critical for the paper to be published.  Most significantly, repeat runs are an absolute must for me: I appreciate that the subfield often does not do these and that the expense of the experiments makes it difficult, but they are a bedrock of the scientific process and something I feel we have to insist on as reviewers and authors to avoid a reproducibility crisis, rather than perpetuating the mistakes of the past.
> > >
> > > b) I still have concerns with how things will behave in finite K regimes and think it is important to either have direct demonstrations the approximations are reasonable in practice, or knock-out end-to-end performance that simply justifies the approach from a fully empirical perspective.  I don't feel like the paper has either at the moment.
> > >
> > > c) The theoretical changes are rather substantial and go beyond what can be reasonably be checked as part of this rebuttal process, while would feel uncomfortable accepting the paper without them being thoroughly checked.  As per b) above, the finite K setting is also what really matters so, though they are useful, I also do not see them as a game-changer for the strength of the work.
> > >
> > > Thus, though I would certainly encourage you to keep working on improvements and resubmit the paper, I think pushing it through now would be a step too far and it needs to go through another review cycle once all the improvements are finished.  I wish you the best of luck with this should the paper indeed be rejected, as I do think it is a promising project, just not quite there yet.

---

### Official Review · AnonReviewer3 · 2020-10-27
**Good attempt but more comparisons with other information theoretic objective functions will make it better.**

**Rating:** 5
**Confidence:** 3

**Review:**

Summary:

This paper proposed a lower-bound on the mutual information by introducing a conditional mutual information term.
The authors claim that their conditional mutual information term can avoid capturing redundant information shared among different data samples and therefore is more efficient.


Reason for the score:

I think the authors could improve their paper by comparing with more methodologies such as the information bottleneck (IB), and various approximations such as MINE from ICLR 2018 (https://openreview.net/pdf?id=rJHOuiqaf).



Pro:
- The authors provided experiments in vision, dialogue, and synthetic datasets on different models.

Con/Questions:
- Some experimental setups are not too clear. For example, how did the authors choose information-restricted views?
- I think the authors could compare their method with other relevant information theoretic methods such as the Information Bottleneck (IB). Many recently applications in vision, language, and molecules found IB very useful in balancing between compression and information retention.
- I think the paper is a bit difficult to follow in its current format particularly with its current choices of words. For example, the authors use both $x, y$ to denote data features instead of data features and data labels which is quite uncommon. "views" is also used frequently in this paper which I can only assume that the authors meant "data samples".

-----------------------------------------------------------------
Post Rebuttal:

Many thanks for the authors to update their original paper addressing some of my questions and concerns.
I have now updated the score.

---

> ### Author Response · Authors · 2020-11-18
> **On IB and MINE and clarifications**
>
> Thanks for your review. We’ll try to clarify hereafter your questions as best as we can:
>
> **About info-restricted views**
>
> In Figure 1, we specify that information restricted views can be obtained either by masking pixels in an image or by omitting past sentences in a dialogue for example. In our vision experiments, we ablate two ways of generating information restricted views (see Table of results). The first is by using “cutout” (e.g. masking out square pixel regions in an image), the second is by using “multi-crop” (e.g. cropping aggressively an image and resizing to the original size). Both strategies entail a loss of information and therefore an “information-restricted” view. A description of this is in Paragraph 5.2. For our dialogue experiments, we consider the last sentence in the past of a dialogue as the information restricted view for the past in the dialogue. This can be found in Paragraph 5.3.
>
> **About IB & MINE**
>
> Although seemingly related, IB is orthogonal from the purpose of this paper: IB tries to minimize MI between X and latent representation Z while maximizing information between Z and labels Y. Therefore, usually it has been used in the context of supervised learning, where the goal is to regularize the model in order to exploit only the information necessary to solve the task. On the contrary, here we want to maximize MI between X and Z with the hope that imbuing Z with the most information about X can help a variety of downstream tasks (e.g. effective pretraining). MINE is a different way of maximizing MI between X and Z. Please refer to (Djelm et al.; 2019) https://arxiv.org/abs/1808.06670 for a comparison between learning representations using MINE and InfoNCE (as is the case in our paper). Current SOTA self-supervised representation models use InfoNCE bounds (that have a low-variance) and therefore we build upon those bounds. Extending our results to MINE-type bound could be interesting future work
>
> **About "views" terminology**
>
> We used y for the other example because in our setting we don’t have any labels as we perform self-supervised learning. We can change the notation to another letter if you think it's clearer. What would you suggest? “views” is a common word in self-supervised learning techniques, e.g. see CMC for example (https://arxiv.org/abs/1906.05849), and slightly differs from data samples in such it is used to describe controllable transformations of data samples (such as rotation, color jittering for images for example).

---

### Official Review · AnonReviewer1 · 2020-10-28
**good work with reasonable idea, needs to clarify its connection with another work.**

**Rating:** 6
**Confidence:** 3

**Review:**

This work proposes to decompose mutual information estimation into subtasks, based on the chain rule of MI, i.e. Eq.(6).
A novel bound for the proposed decomposition insight is achieved based on the classic InforNCE bound. Higher total information is expected to be captured with the proposed method.


The proposed insight is understandable and reasonable to achieve the desired goal. it is
 also presented to be effective to achieve higher MI and results in better representations for classification.
I have one question regarding this work.

1) Work [1]  also presents analysis on estimating mutual information in decomposing, or hierarchy, manner, i.e. Eq(19) in its paper. Can you discuss this work with yours? Can I understand your work as a variant of this work performed with InforNCE estimation? I appreciate your opinion here.


[1] Gao, Shuyang, et al. "Auto-encoding total correlation explanation." The 22nd International Conference on Artificial Intelligence and Statistics. 2019.

---

> ### Author Response · Authors · 2020-11-18
> **Regarding CorEx**
>
> Thanks for the link to the CorEx paper!
>
> The theory in our paper shows how to maximize MI between potentially high-dimensional variables by decomposing it into a sum of potentially smaller conditional MI terms between subsets of such variables, i.e. I(x, z) = \sum_i I(x_i, z | x_<i). Taking on the notation of the CorEx paper, our paper gives a way of maximizing terms such as I(x_i, z | x_j) or I(z_i, x | z_j) which seems outside of the scope of the CorEx paper which aims to maximize \sum_i I(x_i, z) independently. In fact, it seems that CorEx encourages each x_i to be independently informative about z while minimizing the MI between x and z. This will have the effect of maximizing the MI between x and z under some independence assumptions that will have of course some effect on I(x_i, z | x_j), but to which extent is not totally clear to us. We think that our method could be used to measure the magnitude of such conditional MIs for example in CorEx trained models. We can have a similar argument for Eq. 19, where our method could be used to explicitly maximize I(x, z^2 | z^1), but I think this is outside the scope of the CorEx paper.
>
> For the reasons above, we don’t see our paper as a variant of the CorEx paper, but you are correct in saying that CorEx paper maximizes I(z_i, x) using a variational lower-bound, while we generally operate using contrastive lower-bounds on MI. From a very general standpoint, in our paper, we are not concerned with achieving disentanglement of the learnt representations therefore we feel that the general goals of the papers are different but we’ll make sure to include the paper in our references.

---

### Author Response · Authors · 2020-11-18
**General Response, thanks to all reviewers!**

We thank all reviewers for their feedback.

We were glad to see that reviewers generally acknowledged the novelty of the approach as well as the general potential of the idea, the ability of the approach in order to obtain better estimations of large amount of MIs and the fact that experiments span both vision and language setting.

We substantially updated our paper to provide stronger guarantees for our theorems as well as a completed synthetic study on the effectiveness of the importance sampling approximation for conditional mutual information estimation, and will do our best to address each individual concern in the individual responses.

---

### Decision · Program_Chairs · 2021-01-07
**Final Decision**

**Decision:**

Reject

**Comment:**

The authors propose a learning approach based on mutual information maximization. By considering a view x, and two subviews, x’ and x’’, the authors provide a bound on MI by combining two InfoNCE-like bounds on I(x’’; y) and I(x’; y | x’’). The authors show that optimising this (approximate) bound leads to improvements in several tasks covering NLP and vision.

This paper is aiming to address a significant problem for the ICLR community and provide a novel solution. The manuscript is well written and the main idea is clear. The reviewers appreciated the fact that the experimental setup covers both vision and NLP. On the negative side, the reviewers raised several major issues, both with the presented theory and the experimental setup. From the theoretical point of view, the approach hinges on a good  approximation for p(y|x''), which could arguably be as hard as the original problem. The author's response is definitely a step in the right direction, but the changes to the original manuscript are quite substantial and there is no time for a thorough validation of the updated claims. I will hence recommend rejection and strongly suggest that the authors incorporate the reviewers’ feedback and submit the manuscript to a future venue.